# Enhancing Sustainability in Building Design: Hybrid Approaches for Evaluating the Impact of Building Orientation on Thermal Comfort in Semi-Arid Climates



**Salar Salah Muhy Al-Din** [1,*] **, Hourakhsh Ahmad Nia** [2] **and Rokhsaneh Rahbarianyazd** [2]

[1] Faculty of Architecture, Girne American University, N. Cyprus Via Mersin-Turkey, 99320 Kyrenia, Cyprus
[2] Faculty of Architecture, Department of Architecture, Alanya University, 07450 Alanya, Turkey; hourakhsh_ahmadnia@yahoo.com (H.A.N.); rokhsaneh.rahbarianyazd@gmail.com (R.R.)
* Correspondence: salarsmuhyaldin@gmail.com

**Abstract:** The evaluation of human thermal comfort inside buildings plays a pivotal role in reducing energy consumption and enhancing sustainability in the built environment. The estimation of thermal comfort is based on objective (physical factors) and subjective (psychological factors) aspects. This study aimed to find a hybrid way to evaluate more accurate thermal comfort in the buildings as per their orientations. This study assessed the effect of building orientation on thermal comfort conditions in row houses in semi-arid climates, based on a synthesis of the predictive mean vote (PMV) model and the thermal sensation vote (TSV). For this purpose, row houses were selected in the region of this study. This study concluded that the PMV model calculates a lower thermal comfort level than the TSV method, and that the thermal comfort demand within the houses was higher than ASHRAE Standard 55. The occupants inside the houses had a lower thermal tolerance. This implied that the residents of these buildings can consume more energy during the summer, typically the harshest season. This study presented new mathematical models for occupants' thermal comfort evaluation in the study region, depending on the building's orientation. In both models, for assessing thermal performance during both the summer and winter seasons, east-facing buildings consistently ranked as the second-best orientation. This suggested that, overall, east-facing buildings can be considered the best choice throughout the entire year in terms of thermal comfort. This study suggested a novel indicator to evaluate the optimum building orientation in the study area in terms of thermal performance.

**Keywords:** building direction; predictive mean vote method; thermal comfort; thermal sensation vote





## 1. Introduction

A comprehensive understanding of thermal comfort conditions affects occupants' satisfaction and influences energy usage, where maintaining thermal comfort is responsible for a big part of energy consumption in any building. People who are dissatisfied with their surrounding thermal environment are more apt to act unsafely, and their capacity for decision making and manual work breaks down [1]. At the same time, buildings consume 40% of the world's total produced energy, and produce almost one over three of carbon dioxide emissions in the world [2,3]. Half of the end-use energy in buildings is applied to maintain thermal comfort through heating and cooling [4]. In Iraq, sixty to seventy percent of the total building's electrical load goes to maintaining thermal comfort [5]. Moreover, in arid climates, increasing urbanization increases the urban heat island significantly [6], and this negatively impacts thermal comfort.

There is a problem in proposing accurate thermal performance solutions for buildings in the design stages, which is the challenge of evaluating thermal comfort in post-occupancy buildings. This is due to the need to consider a set of environmental, personal, and psychological factors. The predicted mean vote and predicted percentage of dissatisfied

(PMV/PPD) index, as a predicting thermal model, is one of the most prevailing international standards to predict building thermal comfort objectively. However, the prediction of this standard has demonstrated differences when compared with the occupants' thermal sensation vote (TSV) in a subjective way [7,8]. A study of 40 occupants in Indian institutional buildings investigated indoor thermal conditions and their influence on comfort. The PMV model indicated a comfort range of 22.46–25.41 °C, with a neutral temperature of 23.91 °C. In contrast, utilizing the TSV's response, a wider comfort range of 23.25–26.32 °C, with a neutral temperature of 24.83 °C, was noted. The PMV model tended to overestimate comfort, while the subjects exhibited greater sensitivity to cold and reduced sensitivity to heat [9]. Through the implication of the TSV with the traditional PMV approach, the modified PMV model can encompass all types of thermal adaptations, including psychological, physiological, and behavioral aspects, while also considering dynamic characteristics. Case studies that have been conducted on air-conditioned buildings revealed that the root mean square errors for the conventional PMV and the proposed modified PMV were 1.24 and 0.13, respectively. Consequently, in comparison to existing methods for thermal comfort prediction, the proposed modification enhances accuracy by at least 65% [10].

This serves to underscore the pivotal research gaps and accentuate the study's distinctive approach to addressing these voids. By placing a strong emphasis on the comprehensive assessment of thermal comfort through both objective and subjective lenses, and by delving into the implications of building orientation in semi-arid climates, this research significantly enriches the current knowledge landscape. Additionally, this introduction now firmly situates this study's innovative hybrid methodology as a critical stride towards advancing sustainability in architectural design, particularly in regions marked by semi-arid climatic conditions.

Many studies have shown that the building style in semi-arid climates can have a significant role in building thermal performance conditions [11]. For example, residents of traditional dwellings have different thermal comfort levels than those living in contemporary dwellings, as they encompass special thermal adaptive behaviors [12].

Several types of research have been implemented in similar climates regarding thermal comfort. A study has been conducted in Baghdad, Iraq, about the influence of the courtyard on the thermal comfort of the occupants. According to the findings, out of the total number of possible occupation hours in houses in Iraq each year, courtyards can provide up to 38% of comfortable hours [13]. Moreover, according to Radha [14], reaching better thermal comfort in the residential buildings in Sulaymaniyah, Northern Iraq, requires a hybrid ventilation system (passive and active) in the summer, especially during the months of July and August. Abdulhameed [15] studied the improvement of thermal comfort inside buildings in the hot and arid climate of Baghdad, Iraq. The buildings' interior thermal comfort was improved by maintaining an appropriate wall and ceiling thickness of 36 cm and decreasing the amount of transparent space by 50%. Another study demonstrated that introverted plan forms and the building coverage ratio are two crucial elements in the climatic design to ensure thermal comfort in Iran's hot and dry climate. The one-sided shape with a northeast–southwest orientation and northeast location, among the evaluated models, has the most hours (2609 h/year) in the range of thermal comfort and the finest plan form [16]. Furthermore, in multi-story residential complexes in the hot and semi-arid climate zone of Duhok, Iraq, the effect of spatial arrangement differentiation on thermal comfort was studied. The ENVI-met program adopted and simulated the PET index, which measures physiologically comparable temperatures. The study's conclusion highlighted the significance of the urban design characteristics reflected through the spatial configuration of the inhabitants' thermal comfort in open areas [17]. These contextual details are vital, as they underpin the rationale for our investigation into the effect of building positioning on thermal comfort in semi-arid regions. While urban environments have been broadly studied, our research narrowed its focus to individual buildings and their orientations.

Our distinctive contribution lies in addressing a significant research gap, specifically examining how the positioning of buildings influences the thermal comfort experienced

within the residences of our study area, an aspect that has hitherto received limited attention. Furthermore, our study introduced an inventive hybrid approach that combines predictive models for thermal comfort with feedback from occupants regarding their thermal experiences. This pioneering methodology aimed to enhance the precision of thermal design guidelines for residential buildings. Importantly, it aligns with sustainability objectives by targeting reductions in energy usage and enhanced building performance in the face of challenging climatic conditions.

To this end, the evaluation of the divergence and convergence of thermal comfort prediction using thermal models and real thermal sensations of residents, with respect to building orientations inside residences in the study area, has not yet been attempted.

This research aimed to find a hybrid approach to enhance thermal comfort evaluations in houses in semi-arid climates. Consequently, this study's main objective was to determine the existing thermal comfort scenario in the houses of the study area concerning the orientation. This was achieved through determining the measurable variables of the PMV/PPD method based on environmental and physical parameters through direct measurements and field surveys and investigating the required data to find the TSV of the occupants through field surveys.

The purpose of this study was to find more realistic methods for evaluating thermal comfort in the residencies in hot and arid climates in the region of this study.

This study raised the following questions: (1) how relevant is the PMV/PPD model (objective way) for the indoor environment of the houses in the study area, and to what extent can occupants' thermal sensation vote (subjective way) develop the thermal comfort assessment?; and (2) what is the effect of orientation on the thermal comfort scenario in the houses of the Garmian Region, Northern Iraq?

This study hypothesized that synthesizing thermal comfort prediction models and the occupants' thermal sensation vote would improve thermal design criteria in the houses in this study region.

For this purpose, case study buildings were selected as a methodology in this study. According to the IOM [18], 98.7% of the people in Northern Iraq live in houses (including row houses, semi-detached, and detached houses), while only 1% live in flats. Hence, this study focused on the houses as the most prevailed residential units in the region and tried to reach a more accurate evaluation of thermal comfort conditions in this residential sector. Cautious case study buildings (row houses) were chosen based on this study's requirement at Kalar in Garmian. The study region is in the southern part of the Kurdistan region of Iraq, classified as semi-arid, and according to the Köppen–Geiger climate classification system, it has been identified as subtropical steppe (BSh) [19]. The critical season in this area is summer, as it is longer, the temperature arrives at 50 °C in warm months, and the warmer season is longer than the cold season [11,20].

Overall, given the increasing concerns about energy usage and its impact on the environment, it is crucial to prioritize building practices. Buildings are contributors to energy consumption and carbon emissions worldwide, so it is important to assess how building design affects occupants' comfort in terms of temperature. This study addressed this pressing issue by exploring the influence of building orientation on comfort conditions in semi-arid climates, and also made sure to align with sustainability goals by reducing energy usage and optimizing building performance under challenging climatic conditions. By combining subjective and objective assessment methods, this study introduced mathematical models that refine thermal comfort evaluation and provide valuable insights into achieving optimal building orientations for better thermal performance. Through its analysis, this research makes a contribution towards promoting sustainability in building design, especially in regions characterized by semi-arid climates. This is through suggesting guidelines for the optimum orientation, which can help maintain comfortable indoor temperatures by minimizing exposure to extreme outdoor conditions, leading to reduced reliance on heating and cooling systems. By aligning with the local climate and environment, well-oriented

buildings contribute to sustainability by reducing energy consumption and greenhouse gas emissions.

## 2. Materials and Methods

### 2.1. Thermal Comfort

Thermal comfort is challenging to describe, as several environmental and personal variables must be considered. Fanger described thermal comfort in relation to physiological factors when he asserted that thermal comfort is a person's perception that is dependent on the physiological strain placed on him by the environment [21]. Givoni noted that human thermal reactions are not just a function of environmental factors and defined thermal comfort as the absence of annoyance and discomfort because of a cold or heat [22]. The most international definition of thermal comfort is the definition of ASHRAE Standard 55 [23], which is "Thermal comfort is the condition of mind that expresses satisfaction with the thermal environment and is assessed by subjective evaluation" [24]. There are two classifications of factors to consider when thermal comfort is studied: objective factors and subjective factors [25]. Objective factors include air temperature and velocity, relative humidity, mean radiant temperature (radiation), the activity level of the people, and people's clothes [26,27]. Examples of subjective factors include the non-measurable factors like people's psychological and contextual factors, such as previous experience, thermal expectations, time of exposition to surrounding environmental conditions, acclimatization opportunities, cultural factors, and social factors [25,26,28]. Moreover, the orientation of the building affects the ability to collect solar irradiance incidence and the parts that are affected by winds [29,30].

Accordingly, there are several definitions for 'thermal comfort,' and all references to the previous models are based on the environmental and personal factors to achieve thermal comfort [31]. Hence, thermal comfort must be studied within objective (the physiological and physical) and subjective (psychological) aspects to reach a more realistic evaluation of thermal comfort. Therefore, the current study determined the physiological and psychological elements through questionnaires and developments and determined the physical aspects through direct measurements and questionnaires. This study tried to synthesize thermal comfort prediction through the objective models, depending directly on the measures and observations. It combined them with the subjective models obtained from questionnaires and field surveys. This suggested that a new model could reach a more realistic thermal comfort evaluation inside the buildings. The most common objective model is the PMV/PPD model, which is based on heat balance and is commonly used for predicting thermal comfort. On the other hand, the thermal sensation vote is considered one of the standard subjective models [32].

2.1.1. Fanger's Predicted Mean Vote 'PMV' Index and Predicted Percentage of Dissatisfied 'PPD' (Objective Way)

The PMV method, which was introduced by Fanger in 1970, is based on a heat balance model. This method is based on the premise that the effects of the surrounding environment only appear through heat physics and cooperative exchanges between the human body and the environment [33]. Fanger developed a collection of correlations and included them in this model, making the PMV's physical aspect comprise six variables. These four physical aspects are environmental parameters: air temperature, mean radiant temperature (MRT), airspeed, and air relative humidity; the others comprise personal parameters, the level of activity, and clothing resistance [23]. The most applicable international standards, such as ASHRAE or ISO 7730, have employed this method [34], and recommend it to evaluate thermal comfort conditions in naturally ventilated and air-conditioned buildings [23,34]. The PMV method assesses the mean value of many individuals' votes on a heat scale of sensation. This scale comprises seven grads: 'Hot' + 3; 'warm' + 2; 'slightly warm' + 1; 'neutral' 0; 'slightly cool' −1; 'cool' −2; and 'cold' −3. The PMV value should be maintained at the limits of 'neutral' (zero), with a tolerance of ± 0.5 on the seven-point

scale of ASHRAE [23,35]. According to Fanger's analysis of the large size of the data tested in 1970, approximately 5 percent of the population could be dissatisfied even in the ideal situation. Therefore, the PPD was introduced, a quantitative gauge of thermal comfort, as the percentage of a big set of people likely to feel very warm or cold. In the PPD index, individuals with $-3$, $-2$, $+2$, and $+3$ in the PVM assessment would be considered thermally dissatisfied. As mentioned previously, thermal comfort studies have commonly reported that the PMV model cannot always accurately predict the actual thermal sensation of occupants, especially in field settings. According to Hoof [36], this is due to two reasons: the errors are mostly related to the isolation of the clothing and the level of activity. De Dear and Brager [37] checked the behavior of the PMV and found that the PMV overestimates the subjective feeling of warmth of the individuals in warm, naturally ventilated buildings. Humphreys and Nicol [38] also argued that the PMV is only accurate under highly restricted situations for the daily assessment of the comfort vote. The median perceived warmth of warmer surroundings and the coolness of colder environments are steadily overestimated via the PMV. Overestimating the observed warmth can lead to an excess cooling load that is virtually unnecessary. Therefore, using the PMV in the design stage may lead to inaccurate information about the building's actual thermal load requirement to maintain indoor thermal comfort conditions during the summer [38].

### 2.1.2. The Thermal Sensation Vote 'TSV' (Subjective Way)

The TSV can be estimated using questionnaires or direct interviews. This model deals with the psychological and thermal behaviors of the people and the personal perceptions of thermal conditions; therefore, it is one of the subjective models. To assess people's thermal sensation, the participants can select the options on the seven-point ASHRAE-55 scale, and the thermal preference can also be set at the desired time. The answer concerns the thermal environment's acceptability and local thermal discomfort for the inhabitants or building occupants. This method includes using statistics in analyzing the results, and among the used statistical methods is the Likert scale method [39]. Several types of research have studied the variety between the TSV and the PMV [7,8,33,40,41]. Broday et al. [42] asserted that the calculated value of the PMV does not match the TSV obtained via the field survey, and that the PMV model could overestimate or underestimate the TSV of people. However, when the PMV index is compared to the TSV obtained from the field survey, it shows significant contradictions. Rupp and Ghisi [43] addressed that with the (subjective) or adaptive thermal comfort models, the limitations of objective models, like the PMV, became obvious.

### 2.1.3. Building Orientation Effects on Thermal Comfort

The main objective of passive design is to increase or even eradicate the need for active systems, while preserving or improving residents' comfort [44]. Building orientation is a critical element of passive design strategies in arid or semi-arid climates, affecting the ability to collect sun rays and control the effects of winds on the building [29,45]. In the hot season, a western-oriented building is usually more accessible to solar radiation, receives less shading, and is warmer throughout the day; the same situation arises for southern facades, while lower sun rays would be collected by northern-oriented buildings [22]. Hence, the orientation of the building has a considerable effect on the heating and cooling loads, which affects energy consumption in the building [46]. These fundamentals are general, and there are some exceptions according to several climatic and geographical characteristics. Therefore, investigating the effects of the orientation on thermal comfort is essential to reaching an optimum environmental design inside the buildings.

In some climates, like hot and dry or cold climates, buildings must be oriented according to the sun ecliptic. In humid climates, where comfort is basically obtained via air movement, buildings must be oriented towards the predominant winds [47]. Therefore, the main building facade is preferred to be oriented to the northern–southern, where the sun rays in the warm season enter marginally through the facades and openings in those

orientations, as during the winter, when the sun altitude is lower, the sun ray accessibility is higher [48].

*2.2. Methodology*

This methodology proposed the hybrid way of the measurement of thermal comfort based on an objective method (PMV) and a subjective method, the occupant's TSV, the strength of their interrelationship at the essential orientations in buildings, and, in turn, the relation to the climatic characteristics of the study area under extreme seasons throughout the year. As explained in the following parts, many steps have been considered to achieve this methodology.

2.2.1. Criteria for Selecting the Study Samples

Houses contribute a significant part of the dwelling sector, which will consume almost 67% of total energy consumption in the building sector by 2030 around the world [4]. Therefore, thirty-two air-conditioned row houses at New Kalar Community in Kalar, the Garmian Region, have been selected as case study buildings as a dominant category of dwelling types in the region of study by 65.7% of total dwellings [49]. The study area is located in southeast Kurdistan, Iraq, and is specified as subtropical steppe (BSh), according to the Koppen classification. The buildings were distributed in four main orientations (south, north, east, and west), with eight houses in each direction. The selected sample of buildings was chosen carefully to meet the following criteria: the case study buildings must be divided equally into four main orientations, the location of the buildings should be similar to ensure the same micro-climatic characteristics, the building construction materials and the color of the outer surfaces should be the same, the selected buildings should have similar ages, and the case study buildings must have the same form, function, design, and prototype (see Figure 1).

All the residents of the selected buildings have spent similar times in their houses to ensure a similar thermal experience. Age and gender, as the influential factors in thermal comfort [50], were considered during the case study building selection, which made the task more difficult. It must be mentioned that the metabolism is slightly lower with age and gender. Both older men and women prefer almost the same thermal environments. Many types of research have shown that although older adults have a slower metabolism, it is balanced with low evaporation losses [51,52]. Therefore, the age of the residents' sample of the survey was limited to be from 25 to 54 years. The number of occupants in one unit affects thermal comfort evaluation, where each individual releases 80% of the total amount of the produced energy as heat, which is dissipated, and only 20% is used by the human, and which increases the heat inside the building [47]. Hence, the number of occupants in each house was limited to be from 3 to 5 people. Moreover, to achieve gender equity in this study, the number of participants in the questionnaire process was the same for males and females (one female and one male from each house). The reason for adhering to this was to balance the male and female sensation differences [35,53]. It should be stated that within these limitations, the convenience sampling technique was applied during sample selection due to the nature and conditions of this study. Data collection from the case study buildings were implemented daily during the summer and winter; during July and August of 2017 (summer), and January and February of 2018 (winter). Case study buildings' descriptive data were collected via field observations and documentary records as plans and 'cad' files. The PMV/PPD method was employed during this data collection process using field observations (direct measurement). In the same context, TSV data collection was conducted through a field survey (questionnaire). However, the information from both methods were collected simultaneously.

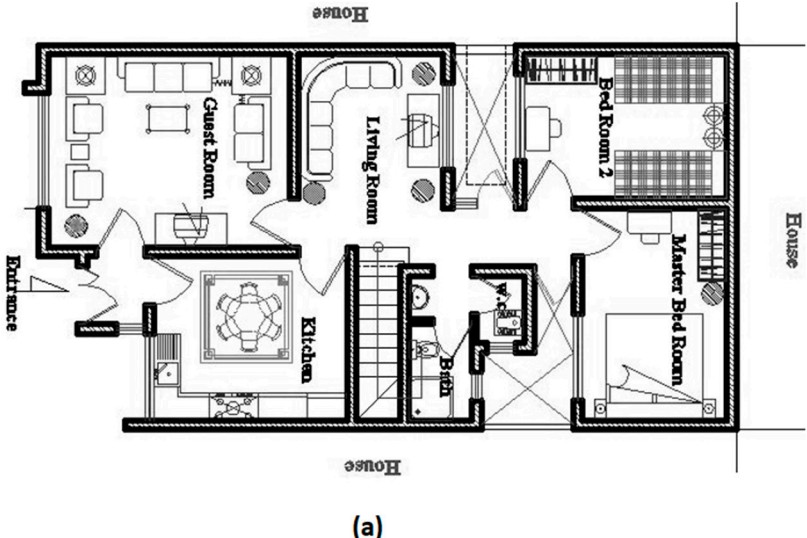

**(a)**

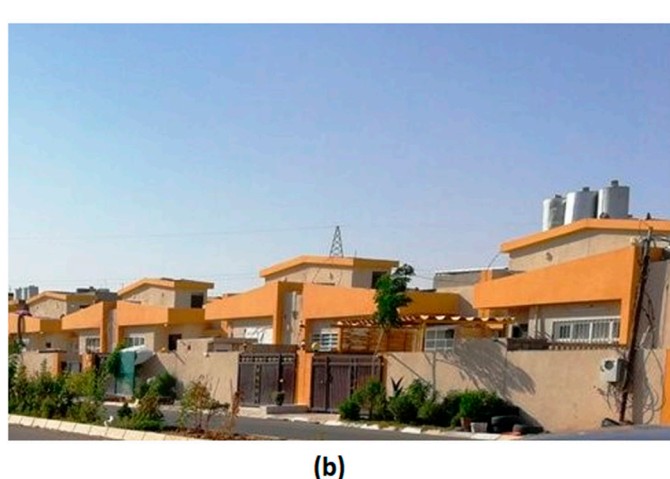

**(b)**

**Figure 1.** (**a**) Selected row houses plan. The building area is 109 m$^2$, and it is a one-floor building. It contains two bedrooms, one living room, one guest room, plus a kitchen, bath, and toilet. (**b**) The shape of the row houses from the outside.

### 2.2.2. PMV/PPD Calculation

1.  In this study, the PMV/PPD was calculated by collecting the environmental aspects (in and out temperature, relative humidity, air movement, and global temperature) to obtain the MRT. It can be calculated using either the CBE thermal comfort tool, as will be introduced later, or in terms of the air temperature (*Ta*), the globe temperature (*Tg*), and air velocity (*Va*) using Equation (1) [54,55]:

$$MRT = \left[ Tg \times \left( 1 + 2.35 \times Va^{0.5} \right) - \left( 2.35 \times Ta \times Va^{0.5} \right) \right] \tag{1}$$

The air temperature was measured from the electronic weather station 'HAMA', and the globe temperature (*Tg*) was measured using an 'EXTECH-H30′ Glob thermometer (40 mm diameter ball size), while an anemometer DA02 model (TACK LIFE) was used to measure the air velocity, as shown in Figure 2.

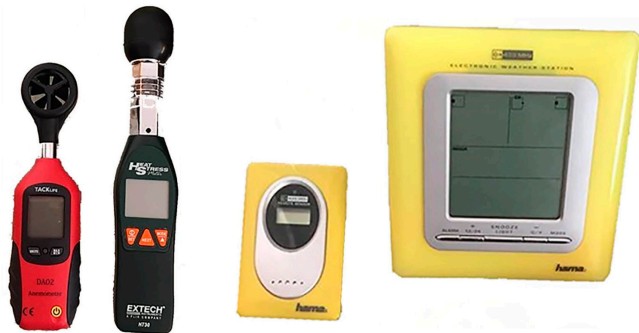

**Figure 2.** The measurement tools for calculating the MRT and PMV/PPD (Hama—electronic weather station located above, EXTECH-H30, Teledyne FLIR Company, Wilsonville, OR, USA, and anemometer DA02-, Tacklife Company, New York, NY, USA).

It should be mentioned that there were slight differences in the MRT outcomes when using both ways; this was due to the effect of the ball size diameter of the globe temperature instrument when it was changed from 150 mm to 40 mm. However, it did not significantly affect the final results of the PMV/PDD predictions [56,57].

2. Occupants' variables data (clothing insulation and metabolic rate) have been obtained through a field survey (a questionnaire form) as required data to calculate the PMV/PPD.

3. Due to the complexity of the equation in obtaining the PMV through conventional calculation methods, several computer tools and programs are available to implement the calculation with several levels of accuracy. In this study, the 'CBE Thermal Comfort Tool' was applied in calculating the PMV/PPD for different orientation buildings in the selected case study. The reason for this was to increase the credibility of the calculation by attaining actual data from direct observation.

The CBE-version 2.0. is an internet application, whose first edition was released in 2014 [58]. The user can choose the required calculations in this software based on ASHRAE or EN Standard [59]. Achieving thermal comfort in an environment is a heavy task. Still, the ASHRAE 55 Standard provides the different equations and tables, thermal environment surveys, and the documentation sample compliance to abide by the standard as simply as possible [60]. Table 1 shows the tools and factors that have been used in the process of predicting thermal comfort level.

**Table 1.** Factors and tools used to evaluate thermal comfort based on the PMV/PPD index.

| No. | Parameter | Tool Model | The Location | Clo Summer | Clo Winter | Met |
|---|---|---|---|---|---|---|
| | Environmental factors | | | | | |
| 1 | Globe temperature ($Tg$)—40 mm diameter | Glob thermometer (EXTECH-H30) | In the building | | | |
| 2 | Air temperature ($Ta$) | Weather station 'HAMA' | In the building | | | |
| 3 | Relative humidity (RH) | The electronic weather station 'HAMA' | In the building | | | |
| 4 | Air velocity ($Va$) | DA02 model anemometer | In the building | | | |
| | Personal factors | | | | | |
| 6 | Insulation of clothing (Clo) and the metabolic rate (MET) | | Bedrooms | 0.5 | 1.03 | 0.8 |
| | | | Kitchen | 0.5 | 1.03 | 1.6 |
| | | | Living room | 0.5 | 1.03 | 1.6 |
| | | | Guest room | 0.5 | 1.03 | 1.0 |

Accordingly, the calculations in this study were arranged in compliance with the ASHRAE 55 Standard as the most appropriate international standard for the current study and were pertinent to the field survey data.

### 2.2.3. TSV Evaluation

The questionnaire form was developed by the authors and distributed to sixty-four occupants, considering the gender factor in the thermal sensation process. The ASHRAE Standard suggests that the response rate on the survey must surpass 35 percent when the participants are more than 45. The minimum number of participants must be 15 if that number is between 20 and 45. However, if the number is less than 20, at least 16 individuals should participate in the questionnaire to make the sample representative [23]. Based on the previous standard, sixteen occupants from each orientation group of buildings voted about their sensations. The total male participants over 40 years old were 25%, and the other 75% were between 25 and 40 years only, while the females above 40 years formulated only 9%, and the remaining 91% were between 25 and 40 years. They were asked to record their thermal sensation in each part of the house simultaneously with the direct observation process for collecting the required data to assess their predicted mean vote (PMV). The questions entailed the participants to evaluate their thermal comfort inside the houses according to their thermal experience. The thermal sensation vote is rated on ASHRAE by '7' points (i.e., −3 cold, −2 cool, −1 slightly cool, 0 neutral, 1 = slightly warm, 2 = warm, and 3 = hot). Therefore, based on the previous standard, this study determined the thermal sensation votes by voting within the seven types of thermal sensation scale to be applied in Likert's formula. The scale was classified as: cold = 7; cool = 6; slightly cool = 5; neutral = 4; slightly warm = 3; warm = 2; and hot = 1 [61]. The direct measurement times were considered the adaptation time for the occupants to fill out the questionnaire in each part of the building when moving from one part to another.

## 3. Results and Discussion

This study is part of a wider study, with more comprehensive findings. Since HVAC systems can affect the results of the thermal assessment, it was necessary to show the HVAC system in each orientation. The average HVAC system applied inside the buildings for different orientations was obtained based on in-situ observations and the questionnaire. The results demonstrate differences in these HVAC systems based on each orientation, as shown in Table 2.

**Table 2.** The average usage of HVAC systems during the summer and winter for different orientations of houses.

| HVAC System | North Orient. | East Orient. | South Orient. | West Orient. |
|---|---|---|---|---|
| **Summertime** | | | | |
| AC cooling | 55% | 70% | 75% | 75% |
| Water cooler | 65% | 55% | 55% | 50% |
| Electric. fans | 73% | 65% | 58% | 53% |
| **Wintertime** | | | | |
| AC heating | 73% | 55% | 52% | 62% |
| Kerosene heater | 15% | 12% | 10% | 12% |
| Elect. heater | 100% | 100% | 100% | 100% |

These results demonstrate the heavy dependence on electrical ways of heating and cooling in the region of this study. The study area witnessed severe weather during summer; therefore, natural ventilation was not considered, as it is not effective in the hot season.

### 3.1. The Results of the PMV/PPD Method (Objective Way)

The results from the objective way of predictive thermal comfort based on the PMV/PPD method were calculated for every eight houses in each orientation. The results of the PMV for each group of buildings in each orientation were identified.

To understand the validity of the obtained results and the confidence level of the groups of the results, the one-way ANOVA test was applied. Nevertheless, to obtain a 95% confidence level, the alpha level value was determined as $p = 0.05$. It was used for every eight groups of the mean radiant temperature inside the houses in each orientation, which was obtained from the direct observation of environmental parameters. The one-way ANOVA test was used to determine whether there are any statistically significant differences between the means of three or more independent (unrelated) groups, called the null hypothesis. If the F-value found in the test is lower than the F-critical value, then the one-way ANOVA results show that it failed to reject the similarity hypothesis. According to the one-way ANOVA test results for all the data groups in each orientation, the similarity was valid, and the null hypothesis was approved in both seasons, as shown in Table 3.

**Table 3.** One-way ANOVA test results demonstrate F- and F-critical values for each orientation group of results.

| Season | Orientation | F-Value | F-Critical Value |
|--------|-------------|---------|------------------|
| Summer | South | 0.015 | 2.313 |
|        | North | 0.027 | 2.313 |
|        | East | 0.022 | 2.313 |
|        | West | 0.024 | 2.313 |
| Winter | South | 0.178 | 2.313 |
|        | North | 0.065 | 2.313 |
|        | East | 0.059 | 2.313 |
|        | West | 0.026 | 2.313 |

Accordingly, the average of the results of the group of PMV/PPD results in each orientation was considered for each orientation. Therefore, the thermal comfort conditions inside the houses were found for the winter and summer in each orientation. In the summer, the average thermal comfort prediction via the PMV/PPD model demonstrated variability concerning the orientation. The southern-oriented houses witnessed warm zones, and the warmer thermal condition indicated via the PMV method was found in the kitchens and guest rooms, rated 'warm.' It should be mentioned that the cooler places were in the master bedrooms and living rooms and were recorded as 'neutral'. The highest PPD was 76%, and the lowest was 5%. The cooler buildings in the summer were the ones that oriented to the north. The 'slightly warm' condition in the kitchens was the warmest thermal prediction inside the buildings. In this orientation, the master bedrooms were the coolest and registered as 'cool' during the test, and the highest PPD was 65%, while the lowest was 5%. The eastern-oriented houses demonstrated 'warm' thermal conditions in the kitchens, and exhibited the highest PPD of 73%, while the cooler places were the master bedrooms and predicted 'Neutral,' and the same went for the guest rooms and living rooms. The lowest PPD was 5% in the guest rooms and master bedrooms. The warmest houses were the houses that were oriented to the west, where the PMV predictions for these buildings demonstrated that the warmest places inside the buildings were the kitchens and guest rooms. It should be stated that the highest PPD record in all the orientations during the summer season was in the western group of buildings in the kitchens, which was 78%, while the lowest PPD in this orientation was 6%. According to the obtained results, through a predicted mean vote, the worst orientation, in terms of thermal comfort inside the buildings in the study area, was the western orientation, and the best orientation was

the northern orientation. Nevertheless, these results demonstrated that eastern-oriented houses exhibit better thermal comfort performance than western- and southern-oriented ones (see Table 4).

**Table 4.** Average PMV/PPD assessment in the summer for the group of buildings in each orientation.

| Orientation | South | | | North | | | East | | | West | | |
|---|---|---|---|---|---|---|---|---|---|---|---|---|
| Zones | Av. PMV | Av. PPD | Av. sens. | Av. PMV | Av. PPD | Av. sens. | Av. PMV | Av. PPD | Av. sens. | Av. PMV | Av. PPD | Av. sens. |
| G room | 1.92 | 73% | Warm | −0.98 | 26% | Cool | −0.02 | 5% | Neutral | 2.03 | 78% | Warm |
| Living room | 0.38 | 8% | Neutral | 0.07 | 5% | Neutral | 0.48 | 10% | Neutral | 0.37 | 8% | Neutral |
| Kitchen | 1.98 | 76% | Warm | 0.87 | 21% | Slightly warm | 1.92 | 73% | Warm | 1.71 | 62% | Warm |
| Bed. R1 (master) | −0.13 | 5% | Neutral | −1.77 | 65% | Cool | 0.05 | 5% | Neutral | −0.26 | 6% | Neutral |
| Bed. R2 | 1.42 | 47% | Slightly warm | −0.6 | 13% | Slightly cool | 1.22 | 36% | Slightly warm | 1.48 | 50% | Slightly warm |

During the summer, regarding thermal comfort, this study observed that bedrooms (permanently occupied ones) and living rooms were ranked as the most comfortable, while kitchens were the least comfortable. The zones that were adjacent to the outer environment were also reported as being less comfortable.

During the winter, the average PMV/PPD showed differences in the different orientations. The southern group showed that the cooler places were the bedrooms, with 'cool', and that its highest PPD was 92%. The most comfortable places were the kitchens, which were assessed 'neutral' according to the average PMV, and the lowest PPD was registered in these places, whose average was 5%. The average PMV/PPD in the northern-oriented buildings demonstrated that there were 'cold' rated places in both bedrooms, and the highest average PPD was registered in these bedrooms, which was 98%. The lowest average PPD inside the buildings of this orientation was in the kitchens, which were recorded as the most comfortable places, being thermally 'neutral'. Nevertheless, the scenario in the eastern buildings displayed that the most relaxing places were the kitchens and was 'neutral', with the lowest average PPD, which was '9%'. The most uncomfortable places inside these buildings were the bedrooms, which were assessed as 'cool', and the highest PPD was 66%. Similar to the eastern-oriented buildings, the western buildings were predicted as 'cool' in the bedrooms, while its average PPD was higher than the one for the eastern-oriented buildings, and it was 89%. The most comfortable places were the kitchens and living rooms, which were assessed as 'neutral', with a lower average PPD in the living rooms, which was 8%. The prediction scenario demonstrated that the most comfortable orientation in the winter is the western-oriented group of buildings, while the northern orientation was the worst orientation. However, the southern and the eastern ones came at the same level as the second. Despite the lower PPD recorded in the southern ones, even with a higher PPD also being registered in this orientation, it was higher than the one reported for the eastern-oriented buildings. It should be stated that, generally, the zones in the eastern buildings were registered more comfortably than the ones in the southern buildings. A common theme that was observed was that the PPD for the eastern-oriented buildings were better than the ones reported for the southern buildings (see Table 5).

**Table 5.** The average evaluation of PMV/PPD in the winter for the case study buildings in each orientation.

| Orientation | South | | | North | | | East | | | West | | |
|---|---|---|---|---|---|---|---|---|---|---|---|---|
| Zones | Av. PMV | Av. PPD | Av. sens. | Av. PMV | Av. PPD | Av. sens. | Av. PMV | Av. PPD | Av. sens. | Av. PMV | Av. PPD | Av. sens. |
| G room | −0.56 | 12% | Slightly cool | −1.64 | 59% | Cool | −0.84 | 20% | Slightly cool | −0.61 | 13% | Slightly cool |
| Living room | −1.16 | 33% | Slightly cool | −1.62 | 58% | Cool | −1.06 | 29% | Slightly cool | −0.39 | 8% | Neutral |
| Kitchen | 0.09 | 5% | Neutral | 0.22 | 6% | Neutral | 0.44 | 9% | Neutral | 0.45 | 9% | Neutral |
| Bed. R1 (master) | −2.42 | 92% | Cool | −2.86 | 98% | Cold | −1.75 | 65% | Cool | −2.11 | 81% | Cool |
| Bed. R2 | −2.4 | 91% | Cool | −2.86 | 98% | Cold | −1.78 | 66% | Cool | −2.31 | 89% | Cool |

However, in the winter, the comfort zones varied; in the case studies, the bedrooms exhibited higher discomfort and heating energy demands due to limited direct solar radiation. Conversely, the kitchens consistently provided greater thermal comfort across all orientations, attributed to internal heat generation from cooking activities and access to direct sunlight, depending on the orientation. The guest rooms and living rooms also offered less discomfort, primarily due to their direct exposure to solar radiation, except for the guest rooms in the northern-oriented buildings, and for the living rooms, their central positionings were surrounded by other spaces acting as buffer zones.

It is important to mention that the living rooms consistently maintained comfort during both seasons due to their central locations and adjacency to other zones, as illustrated in Figure 1a. This underscores the significance of adopting the buffer zone strategy as a passive design approach for architects and designers to ensure thermal comfort and energy efficiency in this study area.

### 3.2. The Results of the Thermal Sensation Votes—TSVs (Subjective Way)

Occupants' thermal sensation votes were investigated to identify the most comfortable parts subjectively inside the houses concerning the orientation of the building by applying Likert's scale and using the Likert's formula for the evaluation. Hence, sixteen occupants (50% males and 50% females) for each orientation were asked to vote about their thermal feeling in each part inside their houses (kitchen, guest room, living room, master bedroom, and second bedroom) during the summer and winter. The findings have demonstrated the following:

During the summer, for the south-oriented houses, occupants' TSVs demonstrated low thermal comfort in the whole of the houses with this orientation; however, the rating differed from one part to another. The kitchens and guest rooms were the warmest parts and were rated as 'hot' based on the ASHRAE 55 Standard thermal sensation scale [35]. The master bedrooms and the living rooms were the most comfortable places in the houses and were rated as 'slightly warm', while the second bedrooms were 'warm'. However, in the north-oriented houses, the occupants' thermal feeling showed a more comfortable condition. The living rooms and the second bedrooms were felt as 'neutral' by the occupants, and the master bedrooms were rated as 'slightly cool' in the summer. The guest rooms and the kitchens were the warmest parts in these buildings and were rated as 'slightly warm', whereas the living rooms and the second bedrooms were 'neutral'. Regarding the eastern orientation, the guest rooms and the kitchens were the warmest parts and were rated as 'Warm' according to the TSVs of the occupants, while the living rooms, the master bedrooms, and the second bedrooms were evaluated as 'slightly warm'. This indicated that occupants' thermal feelings were of discomfort in all these parts. The occupants in western-oriented houses showed their discomfort feeling in all the parts inside the buildings. The guest rooms, the second bedrooms, and the kitchens were the warmest, and were rated

as 'warm'. The living rooms and the master bedrooms were rated as 'slightly warm', as shown in Figure 3.

**Figure 3.** Occupants' TSVs according to the ASHRAE 55 thermal sensation scale in the summer inside the different oriented houses based on Likerts' formula.

Based on the occupants' TSVs, the worse-oriented buildings in the summer for the study area were the southern-oriented ones, while the best was the northern-oriented buildings. Nevertheless, these results demonstrated that the eastern-oriented houses have better thermal comfort performance than the southern-oriented ones; moreover, the eastern-oriented buildings are slightly better than the western-oriented ones.

During the winter, the occupants of the southern-oriented buildings demonstrated thermal comfort in some parts and slight discomfort in other parts inside the houses. The comfort sensation rated the kitchens, the living rooms, and the guest rooms as 'neutral'; however, the bedrooms were rated as 'slightly cool'. In the northern-oriented houses, the occupants' thermal sensation was comfortable in the kitchens, being rated as 'neutral'. Nevertheless, the master bedrooms and the living rooms were rated as 'slightly cool', and the guest rooms and the second bedrooms were rated as 'cool'. Regarding the eastern-oriented houses, the TSVs inside the buildings were thermally comfortable in three parts (the guest rooms, the living rooms, and the kitchens), which were evaluated as 'neutral', while both bedrooms were estimated as 'slightly cool'. Concerning the western-oriented buildings, occupants' votes showed that the sensation in the kitchens and the living rooms, alongside the guest rooms, were 'neutral'. The TSVs in both bedrooms were evaluated as 'slightly cool' by the occupants based on the sensation scale of the ASHRAE 55 Standard (see Figure 4).

The thermal sensation vote demonstrates that the most comfortable orientation in the winter is the southern-oriented buildings, and that the northern orientation is the worst orientation. The eastern and the western orientation buildings had the same level as the second, with slight advantages for the eastern-oriented buildings than for the western ones. This phenomenon can be attributed to the fact that during the transition between seasons, thermal discomfort and supplementary energy consumption are comparatively less pronounced in comparison to buildings oriented in other orientations; therefore, it is better for the occupants.

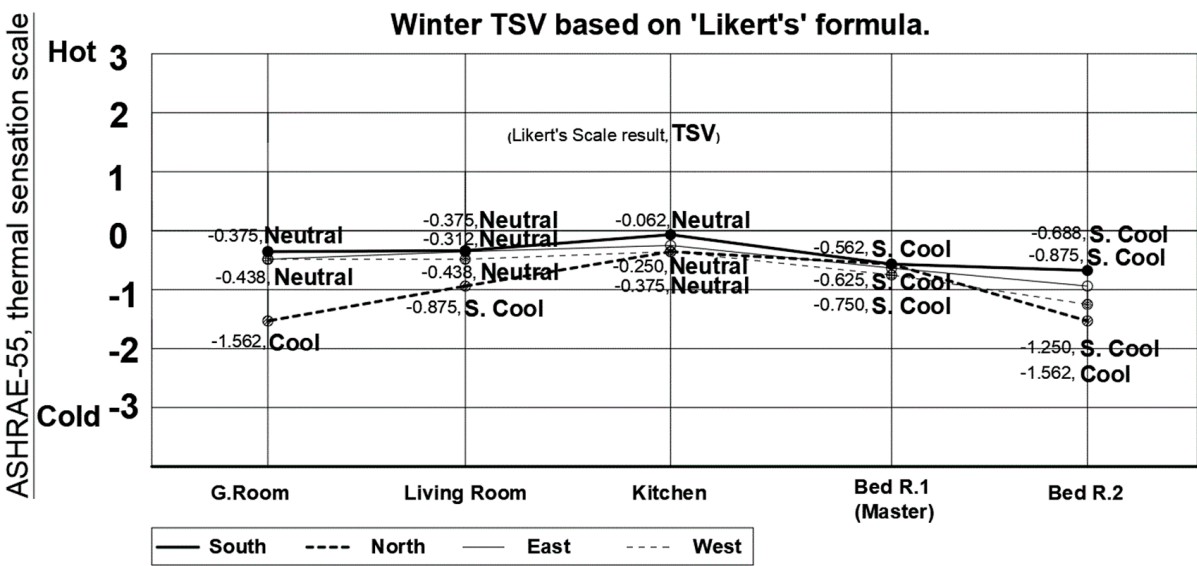

**Figure 4.** TSVs, according to the ASHRAE 55 thermal sensation scale, during the winter inside the distinctly directed houses following 'Likert's' formula.

### 3.3. Discussion

The comparison of the results of thermal comfort evaluation from both objective (PMV) and subjective (TSV) ways demonstrated that the thermal comfort results that were obtained in the summer were relatively harmonized. Moreover, the PMV/PPD method showed more comfortable zones inside the houses in all the summer directions than the TSV method. The matches in thermal evaluation between the PMV and the TSV were occurring in the kitchens as the warmest parts inside the houses in all the orientations, as shown in Figure 5.

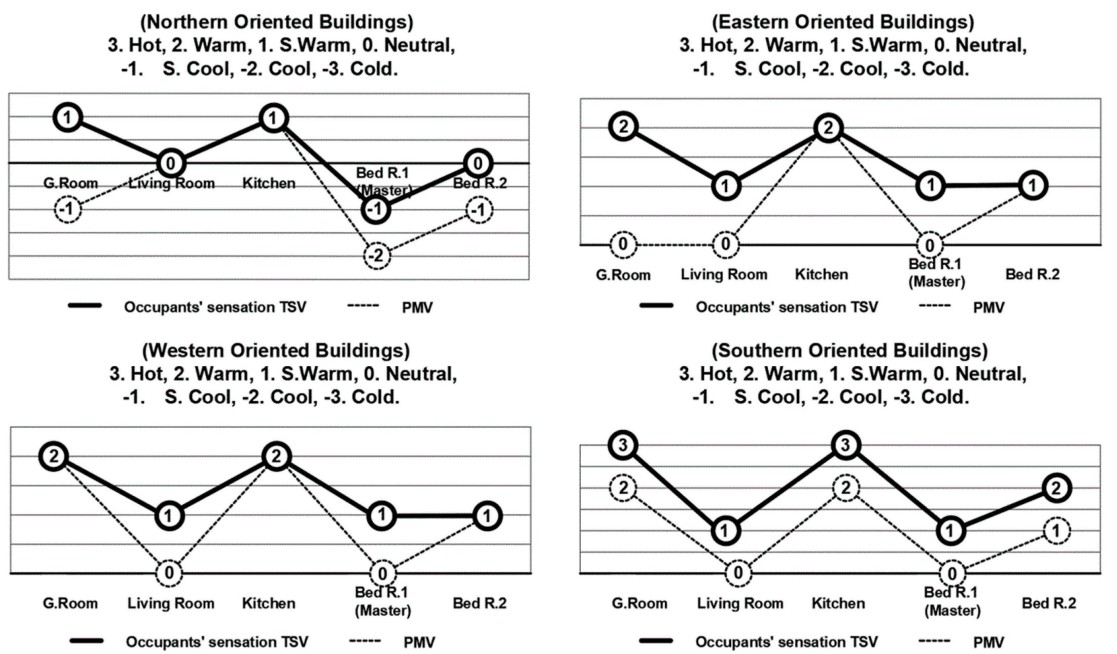

**Figure 5.** The comparison of thermal comfort evaluation in the summer based on occupants' sensation votes (subjective) and the PMV (objective).

During the wintertime, the results demonstrated again that occupants' TSVs were higher than the PMV prediction. Also, the kitchens were the most compatible zones in both methods for thermal comfort evaluation (see Figure 6).

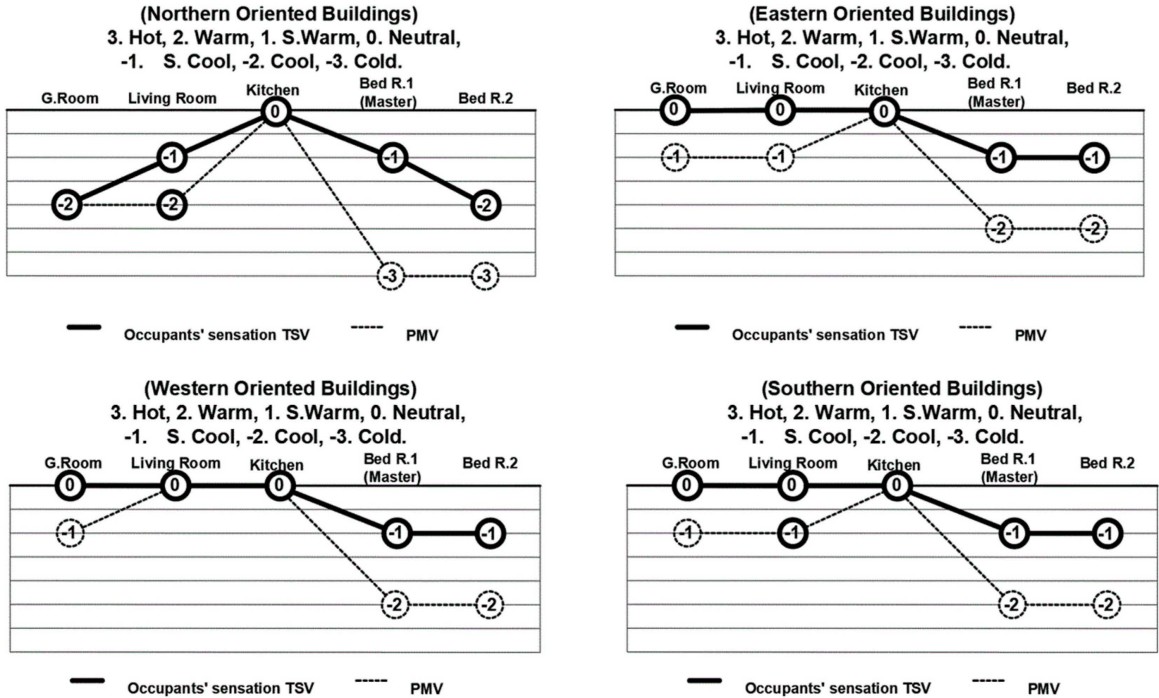

**Figure 6.** The comparison between the results of thermal comfort evaluation inside the case study houses via both the PMV and TSV methods during the winter.

This study found that the thermal comfort demand within the houses was higher than the ASHRAE Standard 55. In other words, based on the selected case studies, occupants inside the houses in the study zone have a lesser thermal enduring. This implies that the residents of these buildings can consume more energy during the summer, typically the harshest season. This indicates that the TSV, or human feelings, dominated the empirical or pure objective ways of evaluating the thermal comfort (PMV/PPD) model based on the outcomes.

The results from both methods were compared and demonstrated that the findings were relatively harmonious in both seasons. Still, the TSV results were commonly slightly higher when compared with the results of the PMV/PPD model. This means that the PMV/PPD model calculates thermal comfort typically lower than the TSV method. These differences have led to some slight changes that made the best and worst buildings' orientations vary in both methods, as shown in Figures 5 and 6. Many factors could affect the dwellers' thermal preferences, like psychological, social, and cultural factors [61]. Contextual elements, such as the individual's thermal history, also have an impact on the building occupants' thermal preferences and expectations [33]. Moreover, a similar study has been conducted in the hot and arid climate of Hyderabad, India, by Indraganti and Rao [50]. They studied various groups of people based on tenure, gender, age, and economic status to assess thermal comfort for the inhabitants. According to this study, the thermal receptions of elderly people and women are higher than that of the thermal comfort standards. Therefore, the PMV/PPD model has been limited to be applied in the context of thermal design, and a number of modifications must be carried out for a more realistic evaluation.

To suggest a model for the relationship between occupants' TSV and PMV in the summertime, linear regression analysis was conducted. The average thermal comfort values without rounding for the groups of results for the TSV and PMV methods were employed to obtain more realistic findings. The correlation coefficient 'r' and the line's

formula was acquired for each orientation in the summer period. Thus, the mathematical correlation models between the objective method (PMV) and subjective method (TSV) were formulated inside the houses of each orientation in the summer. The same process was conducted to formulate the mathematical correlation models between both the subjective and objective methods for the winter period. The results uncovered four mathematical models (one for each orientation) for the summer, and the same number of mathematical models for the winter, as demonstrated in Table 6.

**Table 6.** Mathematical models combine objective and subjective thermal comfort evaluations for the houses during the summer and winter as per the four main orientations.

| Summertime | | |
|---|---|---|
| **Orientation** | **Mathematical Model** | **$r^2$** |
| South | $PMV = 0.9339(TSV) - 0.3686$ | 0.945 |
| North | $PMV = 1.3013(TSV) - 0.6284$ | 0.422 |
| East | $PMV = 0.7126(TSV) - 0.0085$ | 0.197 |
| West | $PMV = 1.677(TSV) - 1.0513$ | 0.950 |
| Wintertime | | |
| Orientation | Mathematical model | $r^2$ |
| South | $PMV = 4.3162(TSV) + 0.4365$ | 0.868 |
| North | $PMV = 1.0771(TSV) - 0.6884$ | 0.224 |
| East | $PMV = 3.1532(TSV) + 0.618$ | 0.721 |
| West | $PMV = 2.8142(TSV) - 0.8352$ | 0.761 |

As it is obvious from the previous table, in the summertime results, the east-oriented houses' tolerance in thermal comfort between the PMV and the TSV was higher than for other orientations, where $r^2$ was 0.197. Nevertheless, in the wintertime, the north-oriented houses demonstrated higher tolerance in thermal comfort between the predicted mean votes and thermal sensation for occupants, where $r^2$ was found to be 0.224.

The above mathematical model was limited to the evaluation of thermal comfort based on the extreme orientations (north, south, east, and west) in the study area, and it was also limited to row houses; a change in the typology of the building may also bring about changes in the results. These results may have displayed different outcomes if extra orientations were considered, as the orientation of the building affects the level of thermal exchange between the building and the outer environment through conduction, convection, and radiation. Moreover, these results were limited to a specific thermal index. The PMV/PPD model was used in this study as one of the most prevalent thermal comfort indices; applying another model could change the outcomes of this study.

The effect of building orientation on thermal comfort was evident in the selected case study buildings. The objective (PMV) method results showed that the buildings facing northwards were the most comfortable in the summer, while those facing west were the least. In the wintertime, the best-oriented buildings were the western-oriented houses, while the northern-oriented houses were the worst. These results revealed minor differences between the PMV/PPD and the TSV, with common harmony in thermal comfort evaluation. However, the subjective (TSV) method's findings showed that the worst-oriented buildings in the summer were the southern-oriented ones, while the best were the northern-oriented buildings. Furthermore, the most comfortable orientation in the winter was the southern orientation, while the northern orientation was the worst orientation. It should be stated that the eastern-oriented houses were always the second-best orientation in both methods for the summer and winter, which should be considered, generally, as the best orientation for the occupants during the whole year in the study area region. This result is a new indicator for the Iraqi Urban Housing Standards Manual, which considers the best

orientation in the study area for the summer and winter to be a southern orientation of 35° east from the south [62]. As a result, a future study using the same methodology to compare the eastern orientation to the orientation mentioned in the Iraqi standards is necessary in order to make an informed decision about changing the standards in the region.

The objective thermal comfort assessment was carried out based on the PMV/PPD model. At the same time, the questionnaire attained the subjective thermal comfort evaluation through the occupants' TSVs. The objective method showed relatively different comfortable zones inside the houses than the subjective method, despite the general harmony in their results.

This section meticulously dissected the research findings, offering a comprehensive analysis of thermal comfort assessment through the objective (PMV) and subjective (TSV) methods. It underscores the alignment between the two methods during the summer and winter seasons, while shedding light on the divergence between them, primarily with the PMV/PPD method predicting generally more favorable thermal conditions within the houses. A notable observation was the revelation that occupants within the study area exhibit lower thermal tolerance, hinting at the likelihood of heightened energy consumption during the rigorous summer period. Additionally, the discussion acknowledges the multifaceted nature of factors influencing occupants' thermal preferences, including psychological, social, and cultural elements, alongside contextual factors like their thermal history. Extensive scrutiny of the comparisons between the PMV and TSV results accentuates the tendency of the PMV/PPD model to yield lower thermal comfort assessments compared to the TSV method. Notably, this study introduced mathematical models that amalgamate objective and subjective thermal comfort evaluations for diverse orientations in both the summer and winter, offering practical insights into their application for achieving more realistic thermal comfort appraisals. Ultimately, this discussion culminates in underscoring the pivotal role of building orientation within the realm of sustainable design and underscores the necessity for future research to evaluate and potentially revise regional housing standards based on these empirical findings.

## 4. Conclusions

This study tried to find a more realistic thermal comfort assessment with respect to the orientations' of the houses of Garmian, Kurdistan of Iraq. The climate of the region is semi-arid characterized by subtropical steppe (BSh), according to the Koppen classification.

The findings indicate that the dwellers have decreased thermal potential. As a result, these residents need more energy to maintain their thermal comfort during the summer, which is frequently the most challenging time of year based on the characteristics of a semi-arid climate. This is because this study found that, while analyzing the thermal comfort (PMV) model, the TSV, or human experience, exceeded the empirical or pure scientific approaches. Thus, the dwellers inside the houses are less thermally tolerant. This suggests that building occupants can use more energy in the summer, which is usually the hottest time of year.

In terms of the effect of building orientation on thermal comfort, the objective model demonstrated that the best thermal comfort conditions in the summer were in the north-oriented buildings, while the worst were those oriented to the west. However, the best orientation in the winter was the west orientation, while the worst was the northern-oriented ones. The subjective model gave different results based on the occupants' thermal sensation, where, despite the best orientation in the summer being the northern orientation, the worst was the southern-oriented buildings. In the same context, in the winter, the best orientation was the south orientation, while the worst was the north. It is worth pointing out that the eastern-oriented houses were always the second-best orientation in both the subjective and objective methods for the summer and winter, which can be suggested as the best orientation in the region of this study.

Thus, this study concluded that the objective thermal comfort prediction might not be sufficient for realistic thermal comfort evaluation without a subjective way to improve

thermal comfort, based on the selected case study houses in the study area. Hence, synthesizing these objective and subjective methods through hybrid mathematical models improves thermal comfort evaluation.

The study presented in this article addresses a significant research gap within the realm of sustainable building design, particularly under semi-arid climates. Its innovation lies in its comprehensive approach to evaluating thermal comfort within buildings, taking into account both objective and subjective factors. While previous research has predominantly concentrated on objective models like the PMV method, this study introduced a hybrid methodology that integrates objective and subjective data, specifically the thermal sensation vote (TSV), to offer a more realistic assessment of thermal comfort. Furthermore, this research delved deeply into the influence of building orientation on thermal comfort, revealing that the optimal orientation varies with the seasons. This dynamic perspective on the impact of building orientation on thermal comfort represents a unique contribution to the existing literature. Additionally, this study proposed a number of mathematical models for assessing thermal comfort based on building orientation, thereby enhancing the applicability of design guidelines. In essence, this research bridges a critical gap by highlighting the significance of occupants' subjective experiences in building design and sustainability endeavors, ultimately providing valuable insights for more energy-efficient and occupant-centric architectural solutions under semi-arid climates. Future studies should extend these methods to diverse building types and climates, potentially leading to revised regional standards and more comprehensive findings.

For this reason, this study suggested mathematical models for evaluating thermal comfort by finding the relationship between the PMV methods and the TSV method as per the building orientation, as shown in Table 6. These mathematical models will help the professionals to evaluate thermal comfort inside the houses with more accurate methods to achieve more successful design criteria in future homes. This study's findings underscore a crucial link to sustainability by highlighting the need to prioritize occupants' subjective thermal comfort experiences in building design. This shift towards a human-centered design aligns seamlessly with sustainable principles, potentially reducing energy consumption and enhancing resource efficiency. By incorporating occupants' perspectives, this study offers a path to more resilient and energy-efficient architectural solutions that resonate with broader sustainability goals.

These findings emphasized that in the realm of residential building design and urban planning, a strategic emphasis on the eastern orientation is paramount in the region of study. This involves aligning residential lots to optimize their eastern facade exposure. Such prioritization yields tangible advantages, notably in the form of enhanced thermal comfort and reduced energy consumption. Moreover, for architects and designers, it is significant to adopt the buffer zone strategy as a passive design approach to ensure better thermal comfort zones and low energy consumption in buildings of the study area.

A future study that applies the same methods as the current study to compare the results of the eastern orientation outcome from this study with the results of the 35 degree angle from the south orientation to the east orientation is highly recommended. This specific angle has been indicated in the Iraqi Housing Manual as the best orientation in the region of this study. Therefore, it is critical to make a precise choice for modifying the standards in the region. Furthermore, it is important to apply the same methodology of this study on other building typologies in the region of study for more comprehensive results about thermal comfort inside the buildings. The expansion of such studies may popularize the outcomes of the current study or find any associated features with the current study region. It should be mentioned that the PMV value is greatly affected by the changes in its independent variables (air temperature, mean radiant temperature, air velocity, relative humidity, metabolic rate, and clothing insulation); therefore, the accuracy of the results may change based on the accuracy of the sensors of direct measurement tools. It is recommended to employ the suggested mathematical model and conduct a comparative analysis against established methods for predicting thermal comfort. This

comparison aims to assess the extent of enhancement in accuracy when evaluating thermal comfort within the buildings located in the study area.

**Author Contributions:** Validation, R.R.; Investigation, S.S.M.A.-D.; Writing—review & editing, H.A.N. All authors have read and agreed to the published version of the manuscript.

**Funding:** This research received no external funding.

**Institutional Review Board Statement:** The study was conducted in accordance with the Declaration of Helsinki, and approved by the postgraduate institute in Girne American University, Kyrenia, N.Cyprus via Mersin, Turkey.

**Informed Consent Statement:** Informed consent was obtained from all subjects involved in the study.

**Data Availability Statement:** Not applicable.

**Conflicts of Interest:** The authors declare no conflict of interest.

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
