# Peer review of "Enhancing Sustainability in Building Design: Hybrid Approaches for Evaluating the Impact of Building Orientation on Thermal Comfort in Semi-Arid Climates"

_sustainability, doi:10.3390/su152015180_

Round 1
Reviewer 1 Report
The manuscript entitled ‘Enhancing Sustainability in Building Design: Hybrid Approaches for Evaluating the Impact of Building Orientation on Thermal Comfort in Semi-Arid Climates‘ may potentially comprise an interesting content, but in its current form, it is even hard to review it. Please, revise it according to at least minimal standard quality. At the current form I suggest rejecting it. Please give some respect to the reviewer’s work.
1. The very idea of the table is to put similar information together. Please, avoid repetitive columns and titles in the one table. E.g.: ‘Result by Likert’s method’ in Tables 6 and 7 or PMD and PPD 4 times in Table 5. If it’s not possible with in a form of a table, then put it into a multi-axes diagram.
2. Language usage is sometimes weird. E.g. ‘Ave’ is never used as an acronym to ‘average’. Please check throughout the paper.
3. Subsection titles are poor. Acronyms should not be used in any titles. What do these parentheses mean: ‘(TSV) Evaluation’, ‘The results of (Objective way)’?
4. Presentation of formulas is a nightmare. Has anyone checked what were you submitting?

5. ‘See Figure 2’ is also not acceptable for a scientific paper. It is not an Instagram post.
6. Diagrams are barely readable: e.g. figures 3 and 4. Please check the resolution.
7. If you add a photo, then the backstage should be plain. No reflections, stands, and vinyl floor are allowed: e.g. figure 2.

2. Language usage is sometimes weird. E.g. ‘Ave’ is never used as an acronym to ‘average’. Please check throughout the paper.
Author Response
I appreciate your dedicated time and effort in examining this manuscript. Modifications have been made to Figure 1, and the diagrams, as well as some texts developed as seen in the attached file. Please See the attachment.

Reviewer 2 Report
The study examines the building orientation effect to attain indoor thermal comfort in a particular residential building typology in Iraq. The study is well structured, and I recommend to publish it after minor modifications. Here are my recommendations:
1. The introduction, between lines 48 and 71:
1.1. What do these paragraphs convey to support your argument? For example, but not limited to, reference 15 discussed urban settings while study focused on Buildings. Please show the contact and different points with your intervention.
1.2. Also, showing the juxtaposition with the international literature will be beneficial, to emphasis the (novelty) as mentioned in the abstract (line 23).
2. The keywords should not those in the title, please alter Thermal Comfort and Building Orientation. In line 97 , it seems that the brackets for rowhouse only please revise.
Author Response

Please Also see the attached document.

Reviewer 3 Report
Abstract: Overall, the abstract provides a solid overview of thermal comfort study, but can enhance its impact by offering a bit more context and quantitative data to support the findings. State the main finding in the end of the abstract
Background: Overall, the introduction provides a solid foundation. However, there are few recommendations for improvement:
1. The paragraph relates to various research and findings concerning thermal comfort; nonetheless, it might benefit from the inclusion of specific facts and examples to clarify the arguments being presented. The incorporation of particular statistics or case studies would enhance the paragraph by providing more informative content.
2. This section highlights the significance of the research in relation to sustainability and energy consumption, although it might provide further elaboration on the real-world implications of the research outcomes. In what ways might these discoveries be implemented in practical contexts such as building design or urban planning?
3. It is advisable to take an organisational method by separating the writing into separate sections or paragraphs, each accompanied by clear headings. This structure facilitates reader navigation and understanding.
Method
Figure 1, 3 and 4 - to update the photo/diagram quality to the higher resolution.
Results
The section details up the summer thermal comfort findings for several building orientations. It provides PMV projections, PPD percentages, and room comfort ratings. Understanding results complexities requires this level of information.
2. Although the passage provides a substantial amount of facts, it would be helpful to incorporate an analysis or discussion on the implications of these discoveries. What implications do these findings have for building design and occupant comfort within the region? The inclusion of contextual information or insightful analysis would greatly enrich the discourse.
3. To consolidate the outcomes, it is advisable to incorporate a brief summary or concluding remark that underscores the principal findings and their importance.
4. While the passage presents the TSV results clearly, it could benefit from some interpretation or discussion of the implications of these findings. For example, why might eastern-oriented houses have better thermal comfort performance, and what are the potential design implications?
Conclusion
In general, the conclusion well summarises the research's outcomes, implications, and potential possibilities for further investigation. This publication offers significant contributions to the knowledge and understanding of building design and thermal comfort, benefiting scholars and practitioners in the respective sector. It is recommended to have a brief section related to potential possibilities for further research or questions that may arise as a result of this study. This can offer a contextual framework for the wider research panorama.
The English in the article is generally clear and well-structured. No major comments or error found in the article
Author Response
I appreciate your dedicated effort in examining this manuscript. Enclosed below are the comprehensive responses and respective amendments, which have been highlighted in the resubmitted manuscript. (Please, see the attached manuscript also).

Reviewer 4 Report
Comments:
The manuscript entitled " Enhancing Sustainability in Building Design: Hybrid Approaches for Evaluating the Impact of Building Orientation on Thermal Comfort in Semi-Arid Climates" has been investigated in detail. While the topic addressed in the manuscript is captivating, there are certain issues that the authors should address.
1- Authors should clearly identify gaps and novelty of the research.
2- The Introduction section should be revised to include a more precise and informative review of recent literature.
3- The keywords should be organized in alphabetical order.
4- The "Discussion" section should be added in a more prominent and argumentative manner. The author should thoroughly analyze the findings and discuss them in relation to previous studies, relevant literature, theoretical frameworks, and empirical evidence.
Comments:
The manuscript entitled " Enhancing Sustainability in Building Design: Hybrid Approaches for Evaluating the Impact of Building Orientation on Thermal Comfort in Semi-Arid Climates" has been investigated in detail. While the topic addressed in the manuscript is captivating, there are certain issues that the authors should address.
1- Authors should clearly identify gaps and novelty of the research.
2- The Introduction section should be revised to include a more precise and informative review of recent literature.
3- The keywords should be organized in alphabetical order.
4- The "Discussion" section should be added in a more prominent and argumentative manner. The author should thoroughly analyze the findings and discuss them in relation to previous studies, relevant literature, theoretical frameworks, and empirical evidence.
Author Response

Please, See the attached Manuscript.

Round 2
Reviewer 1 Report
The authors have neither negotiated the reviewer's comments nor adopted revisions accordingly. Moreover, detailed authors' response has not even been provided. I recommend rejecting the manuscript.
Language usage is sometimes weird. E.g. ‘Ave’ is never used as an acronym to ‘average’. Please check throughout the paper.
Round 3
Reviewer 1 Report
Dear authors,
Thank you for the work completed.